# Co-creating community-driven solutions and policy priorities to address antimicrobial resistance through Responsive Dialogues: A qualitative evaluation from Malawi

Henry Sambakunsi[1,2]*, MacWellings Phiri[1], Thomasena O'Byrne[1,2], John Mankhomwa[1], Raymond Pongolani[1,2], Jo Zaremba[2], Nicholas A. Feasey[1,2,3], Eleanor Macpherson[1,2], Deborah Nyirenda[1,2]

1 Malawi Liverpool Wellcome Programme, Blantyre, Malawi, 2 Liverpool School of Tropical Medicine, Liverpool, United Kingdom, 3 The School of Medicine, University of St Andrews, St Andrews, United Kingdom

* hsambakunsi@mlw.mw

## Abstract

Inappropriate antibiotics use due to low public knowledge and awareness contributes to the growing Antimicrobial Resistance (AMR) burden, especially in low- and middle-income countries (LMICs). The aim of Responsive Dialogues (RDs) is to engage communities and key stakeholders to share information, galvanize action and create local solutions to reduce the burden of AMR. RDs promote collaborative problem-solving and have been identified as a potential tool for engaging diverse stakeholders in addressing AMR. However, RDs were only recently developed, and empirical evidence about their utility and feasibility in LMICs is currently limited. We piloted RDs with poultry farmers, government and private pharmacists or prescribers, and male caregivers in Malawi. Using 13 semi-structured interviews and three focus groups, we assessed feasibility of this participatory approach to influence inclusive local AMR policies and solutions. The participatory nature of RDs promoted inclusive decision-making and facilitated participants' capacity to co-create locally relevant AMR solutions. Information shared during RDs improved AMR awareness, prompting participants to recognize and reconsider antibiotic practices and co-create solutions including community education on responsible use, local stewardship committees, and integrating AMR education into existing health programs. Implementation barriers varied: farmers and private prescribers expressed concerns about economic losses from reduced antibiotic use, while male caregivers reported difficulties educating others without tangible resources. Considerations for AMR-RDs scale-up include resource allocation, integration with existing health systems, and further research on long-term impact across different low resource settings. Despite these challenges, this pilot demonstrates that RDs offer a valuable participatory approach for co-creating contextually relevant AMR solutions in resource-limited settings. Sustained

**Data availability statement:** The qualitative data (audio transcripts) underlying this study contain potentially identifiable information about study participants. The ethical approval granted by College of Medicine Research and Ethics Committee (COMREC) (P.07/21/3360) and the informed consent obtained from participants did not include explicit consent for public sharing of transcripts or anonymized data via a public repository. In accordance with these ethical restrictions and to protect participant confidentiality, the data cannot be made publicly available. Relevant data excerpts supporting the findings are included within the paper and supporting information files. Data access requests can be directed to crsu@mlw.mw.

**Funding:** This work was supported by Wellcome Trust (grant number MR/S004793/1) received by EM. The funder had no role in study design, data collection and analysis, decision to publish, or preparation of the manuscript.

**Competing interests:** The authors have declared that no competing interests exist.

impact requires addressing structural barriers, establishing feedback mechanisms, and integrating RDs outputs within existing health system structures and national AMR action plans.

## Introduction

Antimicrobial resistance (AMR) has emerged as a significant global health threat, with particularly severe implications for low- and middle-income countries (LMICs) [1]. The World Health Organization (WHO) has identified AMR as one of the top ten global public health threats facing humanity [1]. Current estimates suggest that AMR was directly responsible for 1.27 million deaths globally in 2019, with the highest burden in sub-Saharan Africa (sSA) [2]. Projections indicate this could rise to 10 million deaths annually by 2050 under worst-case scenarios if left unchecked [3].

In Malawi, where the health system faces resource constraints and the burden of severe bacterial infections remains high, AMR has significant clinical and economic consequences. Recent prospective cohort studies from Malawi have demonstrated that third-generation cephalosporin resistance is associated with increased mortality and prolonged hospital stays, as well as substantial individual and population-level economic costs [4,5]. Surveillance data from the WHO African Region indicates that sSA experiences the highest global AMR mortality rate at 23.5 deaths per 100,000 population, though country-specific comprehensive data for Malawi remains limited [6]. Longitudinal surveillance data from Queen Elizabeth Central Hospital (QECH) in Blantyre demonstrates alarming trends in AMR in bloodstream infection isolates from 1998-2016, with significant increases in resistance to commonly used antimicrobials among both pediatric and adult patients [7]. This trend not only threatens individual health outcomes but also potentially affects food security, economic development, and the achievement of the Sustainable Development Goals (SDGs) [8].

The use and misuse of antimicrobials in both human and animal health sectors is widely recognised as a key driver of AMR [9]. In Malawi, challenges such as inadequate regulation of antibiotic use, limited diagnostic capabilities, and low public awareness combined with a high burden of severe bacterial infection are thought to contribute to the growing threat of AMR [10,11]. A recent review documented similar patterns across sSA, where structural health system challenges compound the AMR crisis [12]. These challenges are further compounded by multi-sectoral and structural factors common in LMICs including weak regulatory frameworks and poor enforcement of existing policies on antibiotics [12]. These gaps facilitate widespread over-the-counter dispensing of antimicrobials, often by unqualified and profit-driven providers [10,13]. In settings where access to formal healthcare is constrained by limited functional capacity, distance, and cost, informal health services including drug shops, drug dispensers, and traditional practitioners become the primary point of care for many communities, serving as patches for structural deficits in the health system [14]. Economic constraints further drive inappropriate antibiotic use: patients save money by bypassing formal healthcare, while drug sellers profit from unregulated

antibiotic sales [14]. In the agricultural sector, farmers in LMICs often rely on antibiotics as a low-cost solution to prevent disease and promote growth in livestock, particularly in the absence of access to veterinary services and support systems [14]. These interconnected structural factors spanning human health, animal health, and regulatory sectors underscore the need for multi-sectoral, participatory approaches to addressing AMR that engage communities and stakeholders across the One Health spectrum.

Recognizing the urgency of this issue, the Malawian government established a Ministerial AMR Unit to coordinate the country#39;s response, culminating in a National Action Plan [15]. The successful implementation of this requires engagement from key stakeholders, including affected community members, to co-create and implement effective interventions [16]. However, there is limited empirical evidence on how participatory approaches function in practice in resource-limited settings, particularly for addressing complex multi-sectoral challenges like AMR.

Responsive Dialogues (RDs) are a participatory public engagement approach, initially developed by Wellcome and more recently revised by the International Centre for Antimicrobial Resistance Solutions (ICARS) in collaboration with the University of the Western Cape [17]. The RDs toolkit uses a participatory approach to bring communities, policymakers, researchers, and civil society groups together to reflect on AMR and the evidence behind it, with the aim of galvanizing action and co-creating local solutions to reduce the burden of AMR. Malawi and Thailand were among the first countries to pilot the toolkit. A recent evaluation from Thailand demonstrated RDs' potential for generating locally relevant solutions, including spontaneous community-led AMR initiatives, though long-term impact assessment remains limited [18]. In addition, while participatory approaches have shown promise in addressing public health challenges [19], there is limited evidence on the effectiveness of RDs in tackling AMR, particularly in LMICs. This study presents a qualitative process evaluation of the RDs pilot in Malawi, examining how participants experienced the engagement process, what solutions they co-created, and what factors they identified as facilitating or hindering implementation.

## Methods

### Ethics statement

Human research ethics approval was received through the College of Medicine Research Ethics Committee (COMREC), in Malawi (Protocol Number: P.07/21/3360) on 15 November 2021, prior to commencement of the research. The study was conducted in accordance with the ethical principles outlined in the Declaration of Helsinki. Written informed consent was obtained from all participants prior to their participation in SSIs and FGDs. Participants were provided with information in both English and Chichewa explaining the study purpose, procedures, potential risks and benefits, confidentiality measures, and their right to withdraw at any time without consequences. All participants were adults aged 18 years and above who had previously participated in the RDs. Participants were reassured that their involvement was voluntary, their contributions would be kept confidential, and participation would not impact their involvement in future workshops or services. Participants were informed that interviews would be audio-recorded as part of the consent process and were advised to avoid using identifying information during recorded sessions.

### Study context: RDs implementation in Malawi

Malawi is a low-income country in sub-Saharan Africa with a predominantly rural population and a high burden of infectious diseases. Limited healthcare infrastructure, widespread antibiotic use in both human and animal health sectors, and constrained regulatory capacity make AMR a particularly pressing public health concern in this context. With Wellcome funding, the Malawi Liverpool Wellcome Programme (MLW) conducted RDs from 2020 to 2023 to enhance inclusive participatory processes to address AMR in Malawi. The initiative piloted the RDs toolkit [14], developed through iterative consultation with stakeholders in two countries including Malawi and Thailand. In Malawi, the AMR RDs were piloted with participants from rural, peri-urban and urban areas in Blantyre in three phases.

**The scoping and design phase.** This involved mapping the AMR ecosystem in Malawi, identifying key stakeholders (e.g., policymakers, researchers, local leaders, media personnel, healthcare providers) and existing AMR initiatives. This was done through consultations and in collaboration with the AMR unit at the Malawi Ministry of Health. Following this process, a stakeholder workshop was convened to set project goals and identify priority participant groups. Three target groups were identified: small-scale farmers using antibiotics to raise animals; private and public antibiotic prescribers; and male caregivers.

**The Conversations Events (CEs).** These were 3 day-long meetings where the identified stakeholders and participants together explored AMR topics and co-created solutions to address AMR. Two CEs were held for each participant group, and specific stakeholders were invited to each event to share expert knowledge of AMR to facilitate informed deliberations. The co-creation activities, however, happened in the absence of experts to prevent power dynamics from influencing group outputs. The implementation period coincided with the COVID-19 pandemic, requiring adaptations to ensure participant safety while maintaining engagement quality. These adaptations included conducting CEs in well-ventilated spaces, implementing mask-wearing protocols, providing hand washing stations and hand sanitizer, and maintaining physical distancing between participants.

**Synthesizing learnings from the first two phases.** Each participant group attended a third workshop where participants, together with the implementation team, invited community leaders, and members of the national AMR unit, to deliberate and consolidate the outputs from the CEs and co-create policy priorities. The outputs generated from all the three participant groups were finally presented to stakeholders at a national policy dialogue workshop to inform the Malawi National Action Plan on AMR.

## Data collection

This qualitative evaluation was conducted between February 2022 and July 2022 approximately 3 months after the final CEs held in November 2021. Data were collected through semi-structured interviews (SSIs) (n = 15) and focus group discussions (FGDs) (n = 3) to capture both individual perspectives and group experiences of the RDs process. SSIs allowed for in-depth exploration of personal experiences and changes in practice, while FGDs facilitated collective reflections on implementation barriers and facilitators. Each FGD had eight participants with equal gender representation (see Table 1), lasting approximately 90–120 minutes.

A topic guide was developed to facilitate discussion about participant's experiences as co-producers of AMR solutions and feasibility of co-created solutions. Broad topics of the guide included perceptions about the RDs toolkit, views on the process of implementing the RDs, experiences of participating in the project including perceived successes and challenges of implementing the co-created solutions. We also asked the participants to identify self-reported changes in knowledge and practices, discuss perceived barriers and enablers to implementing solutions, and participants views on whether the co-produced solutions might be sustained beyond the workshops. These topics were formulated by the research team when developing the research protocol and were iteratively modified throughout data collection. All the interviews and FGDs were conducted in Chichewa by a native-speaking interviewer (HS) and with support for note taking from (RP).

**Table 1. Participants for semi-structured interviews and focus group discussions.**

| Participant Group | Semi-structured interviews | Focus group discussions |
|---|---|---|
| Male caregivers | 5 | 8 |
| Farmers | 5 | 8 |
| Healthcare provider | 5 | 8 |
| Total | 15 | 24 |

## Participants

The study participants included individuals from the three main groups of participants in the RDs namely livestock farmers (cattle and poultry), male caregivers from the community, and healthcare workers including nurses, pharmacists, and clinicians. Cattle and poultry farmers were included because livestock farming in Malawi involves substantial antibiotic use for disease prevention and growth promotion, representing a critical stakeholder group within the One Health framework. The male caregivers were prioritized because they were seen to be less engaged in health research or formal healthcare, more likely to use unprescribed antibiotics, and often responsible for household decision-making.

Participants were purposively selected from RDs attendees to ensure representation across all three participant groups (farmers, healthcare providers, male caregivers) and geographic settings (rural, peri-urban, urban areas in Blantyre). Potential participants were identified from workshop records and invited to participate by the research team based on their availability and willingness to discuss their experiences. Following identification and contact by the research team, participants were provided with the information sheet and consent form and given the opportunity to ask further questions about the study. Participants were reassured that their involvement was voluntary, their contributions would be kept confidential, and participation would not impact their involvement in future workshops or services. The SSIs were conducted where participants found most convenient such as their homes, or places of work such as pharmacies while FGDs were conducted at community halls. For FGDs, ground rules were established at the beginning of each session, including an agreement among participants to maintain confidentiality about what was discussed. Identification numbers were given to each participant to ensure anonymity during transcription and analysis.

## Data management and analysis

All SSIs and FGDs were audio-recorded, transcribed verbatim, and translated into English by RP. Four translated transcripts were randomly checked by MP to ensure data accuracy. The data was then analysed using inductive and deductive thematic approaches. The steps used included initial and ongoing familiarisation with data, development of codes using NVivo 12 Software, including indexing and charting, summarising/interpretation of data, and the development of themes.

To ensure accurate interpretation of the data, quality control measures were put in place. Data were systematically coded by HS, initial coding was reviewed by DN, and developed codes and themes were discussed and further refined by team members MP, JM, RP, EM. Once the preliminary analysis was completed, consultation sessions were held with all researchers.

Quotes are presented to illustrate the findings, with abbreviated identifiers provided for each participant group (HCP = Healthcare Provider; FC = Farmer/ Community member, MCG = Male Care giver).

## Results

Evaluation of our implementation of the RDs approach in Malawi revealed four main themes: 1) Improved participant understanding of AMR, 2) reported and intended changes in practices related to antibiotic use and AMR, 3) Challenges impacting implementation of co-created solutions, and 4) Feasibility and limitations of the RD process.

### Improved participants' understanding of AMR

Data from both FGDs and SSIs emphasized the perceived transformative role of RDs in disseminating information and improving people#39;s knowledge regarding AMR. Most participants reported that they knew little or nothing about AMR before the RDs. The qualitative assessment showed how involvement in RDs transformed their understanding across the various participant groups.

Male caregivers, who initially reported limited knowledge of AMR, gained awareness through the workshops. These participants characterized the dialogues as an eye-opener, with many encountering the term "antimicrobial resistance"

and its public health implications for the first time. The interactive nature of the dialogues allowed them to engage with the information actively, ask questions, relate the concepts to their own experiences, and share new ideas with their peers. This participatory approach appeared to facilitate a deeper understanding of AMR, moving beyond mere factual recall to what participants described as a more comprehensive understanding of the issue. Caregivers reported coming to recognize the dangers associated with AMR and the importance of correct antibiotic use. The realization that common practices, such as self-medication or not completing prescribed courses of antibiotics, could contribute to this global health threat was described as eye-opening. This reported newfound awareness extended beyond the immediate health implications, encompassing an understanding of the broader societal and economic impacts of AMR.

> "At first, I didn't know about Antimicrobial Resistance, but after participating in the conversation events that we had with our colleagues, that#39;s when I gained knowledge that antimicrobial resistance is dangerous and that using drugs without following proper instructions is a bad habit." (MCG, SSI 014)

Similarly, farmers' self-reported understanding of antibiotics and AMR improved substantially, marking what appeared to be a significant shift in knowledge within this crucial stakeholder group. Prior to their participation in the RDs, most farmers reported having limited or incorrect information about antibiotics, their uses, and the potential consequences of their misuse. The RDs were seen to serve as a transformative educational experience, introducing farmers to the concept of antibiotics as specific medications designed to combat bacterial infections. This reported newfound knowledge extended to AMR implications for both animal husbandry and human health.

> "I also want to add that when I was coming to participate in those events, I never knew what an antibiotic was, but after participating in the events, that's when I gained some knowledge on what an antibiotic is and what it does. To me, it was an eye-opener, and we shared new ideas with our fellow farmers." (Participant 3, FC, FGD)

Healthcare providers, despite their professional background, also reported enhanced understanding, highlighting the potential impact of the RDs process. This revelation is particularly significant given that these individuals already possessed a foundational understanding of antibiotics and their use through their formal medical training. The fact that even healthcare professionals reported enhancing their comprehension of AMR underscores the complexity and evolving nature of this global health challenge. For most, their previous exposure to AMR concepts in academic settings had not fully conveyed the scale and urgency of the issue. The dialogues served to bridge the gap between theoretical knowledge and practical understanding, providing a more comprehensive and up-to-date perspective on the AMR crisis.

> "I heard about it from college in the Pharmacology course, but I never knew that the problem is that huge until I participated in the meetings..." (HCP, SSI 07)

### Reported and intended changes in practices related to AMR and antibiotics use

Following participation in the RDs, both FGD and SSI participants' reported intentions to use antibiotics responsibly. Most participants, especially male caregivers and farmers, stated that the RDs helped them to become more cautious about antimicrobials and expressed intentions to refrain from using them without a prescription.

> "Before, I used to think that antibiotics were a cure for everything. But after participating in the RDs and learning about AMR, I realized the importance of using antibiotics only when necessary and as prescribed. I've changed my attitude towards antibiotic use and I'm more cautious now." (MCG, SSI 014)

Public healthcare workers who prescribed antibiotics at the community level reported that they adjusted their antibiotics management practices. Most of them reported a decline in their tendency to collect and keep antibiotics at home for self-prescription. They expressed that they now preferred professionally delivered prescription because they had better understanding of the risks of AMR and the importance of proper antibiotic use.

> "I will do differently especially at home, because I had a bad habit of taking drugs here and store them at home so that if my child gets sick when I'm at work he can take the medicine at home, but now I make sure that he gets prescription first." (HCP, SSI 06)

Participants, including most of the male caregivers, also expressed intentions to promote responsible antibiotic use more widely in their communities. Many reported a sense of responsibility to educate their friends, family, and neighbours about the proper use of antibiotics and the risks of AMR. They described plans to organize community meetings, distribute information materials, and engage in one-on-one conversations to share their knowledge. Some participants even expressed interest in collaborating with local health facilities and schools to reach a broader audience. This reported desire to improve community awareness appears to stem from their realization that AMR directly impact their own health outcomes and those of their families and neighbors, combined with recognition of the critical role that individual actions play in combating it. By taking on this advocacy role, participants hoped to create a ripple effect of positive change, fostering a community-wide culture of responsible antibiotic use that could contribute to preserving the effectiveness of these crucial medications.

> "I feel a responsibility to educate others about AMR and the proper use of antibiotics. I want to organize community meetings and share the information I learned in the RDs. Together, we can make a difference." (Participant 3, FC, FGD)

### Feasibility and limitations of the Responsive Dialogues process

Both FGD and SSI participants reported viewing the RDs process positively, expressing appreciation for the inclusive discussions and competent facilitation. Many participants reported that the inclusive nature of the RDs appeared to allow for a diverse range of perspectives to be heard. They stated that the facilitators seemed to demonstrate a high level of expertise in guiding the conversations, reportedly ensuring that complex topics were broken down into understandable components. Participants described that the depth of information provided, particularly regarding the mechanisms of antibiotic action and the distinction between antibiotics and other medications such as painkillers, was valuable. They reported that the interactive format of the RDs appeared to encourage active participation, seemingly allowing attendees to ask questions, share their own experiences, and engage in meaningful dialogue with both experts and peers.

> "Our interaction was very good and especially on these issues of antibiotics they really went deep on it... They were giving us insights on how the antibiotics work and how painkillers work." (HCP, SSI 07)

The final co-creation event was reported as being particularly well-received by participants. Many described this event as a highlight, bringing together diverse stakeholders from various backgrounds, all united by their shared interest in addressing AMR. Participants reported that the convergence of different perspectives including from healthcare professionals and community leaders to policymakers and researchers, appeared to create a dynamic environment. They reported appreciating the opportunity to engage in intersectoral policy dialogue during this event.

> "The final event was an outstanding event because it brought various people together whereby, we shared ideas and co-created ideas which we thought would be helpful to the whole country." (MCG, SSI 011)

However, participants also reported several limitations and areas for improvement in the RD process. Many participants expressed that more time was needed for discussions, highlighting what they perceived as a significant constraint in the otherwise well-received approach. They reported that this perceived time limitation reflected the complexity and depth of the topics being addressed, particularly in relation to AMR and its multifaceted implications. Participants stated that they found themselves grappling with numerous interrelated issues, from the scientific aspects of AMR to the society and economic factors influencing antibiotic use. They reported that the time constraints often led to a sense of rushing through important topics, potentially curtailing deeper exploration, and thorough understanding.

"The only problem that I encountered was time, we were not having enough time. I'm saying this because we were having several topics to discuss so sometimes due to lack of time we were being rushed to finish one topic." (MCG, SSI 013)

**Challenges impacting implementation of co-created solutions**

Through the RDs process, participants co-created several context-specific solutions to address AMR in their communities. These included community-led awareness campaigns on responsible antibiotic use, local antibiotic stewardship committees, integrating AMR education into existing community health programs and promoting alternative medicine to reduce antibiotic use in livestock. To facilitate community dissemination of these messages, a local cartoonist translated the co-created solutions into culturally appropriate visual materials depicting common scenarios of antibiotic misuse and proper use practices. These cartoon-style illustrations portrayed locally relevant situations to address challenges such as self-prescribing antibiotics, not completing prescribed antibiotics dose, and sharing leftover antibiotics, making the AMR concepts accessible to community members with varying literacy levels.

FGD and SSI participants however expressed mixed views about these co-created solutions. Some reported being enthusiastic about local action, seeing the proposed initiatives as tangible ways to make a difference in their communities. These individuals expressed excitement about the prospect of implementing grassroots campaigns, viewing them as potentially empowering opportunities to take control of a global issue at a local level. They reported appreciating how the solutions were tailored to their specific context, making them seem achievable and relevant. The enthusiastic participants were particularly drawn to the immediate applicability of certain ideas, such as awareness campaigns, which they felt could be launched quickly with minimal resources. Their optimism was fuelled by a sense of collective responsibility and the belief that community-driven efforts could lead to meaningful change in antibiotic use practices.

"The ideas we came up with together, like the awareness campaigns, are things we can start doing in our communities right away. It gives us a way to make a difference." (Participant 2, MCG, FGD)

Despite their reported interest in implementing the co-created solutions to address AMR, many participants also emphasized the need for external support and regulation. While acknowledging the importance of community-driven initiatives, these individuals expressed recognition of the limitations of local action alone in addressing the complex issue of AMR. They argued that comprehensive change would require intervention from higher levels of authority, including government bodies and regulatory agencies. Farmers highlighted the need for stricter enforcement of existing regulations on antibiotic sales, as well as the implementation of new policies to control antibiotic use in both human health and agriculture. They argued that without a robust regulatory framework, community efforts might be undermined by continued easy access to antibiotics through unauthorized channels. Additionally, some participants called for increased financial and logistical support from external sources, such as national health departments or international organizations, to bolster community-driven AMR initiatives.

"The challenges would be there, because if this behaviour of selling drugs in unauthorized groceries continues then this problem won't end. We need strict regulations to control the sale and use of antibiotics. People need to understand the consequences of misuse and overuse." (Participant 8, FC, FGD)

Notably, some participants pointed out that the absence of large-scale poultry farmers in the RDs was a significant limitation, revealing a potential gap in the stakeholder engagement process. Participants suggested that the exclusion of large-scale poultry farmers from the discussions represented a missed opportunity to address one of the major contributors to antibiotic misuse in the agricultural sector. This observation highlights the complex nature of AMR, emphasizing the need for an inclusive approach targeting all relevant actors in the antibiotic use chain. Participants stated that the practices of these large-scale poultry farmers could have substantial impacts on AMR development, potentially undermining the efforts made by smaller-scale farmers and other community members. The absence of these key stakeholders meant that their perspectives, challenges, and potential solutions were not integrated into the outputs, thereby limiting the effectiveness of the co-created solutions.

"However, some people are still misusing antibiotics and most of them are big chicken companies, I'm not sure if they are aware of this issue of antimicrobial resistance because they are still into that bad habit." (Participant 2, HCP, FGD)

Some participants expressed frustration with the lack of feedback on implementation progress. This sentiment appeared to reflect a growing sense of disappointment among those who had invested time and energy in the co-creation process. Many reported feeling that their efforts and enthusiasm had been met with silence or lack of support, leaving them uncertain about the impact of their contributions. The absence of follow-up communication created a perceived gap between the RDs and any tangible outcomes, leading to questions about the long-term commitment to the initiatives they had co-created. This lack of feedback not only dampened morale but also hindered participants' ability to gauge the feasibility of the proposed solutions or share experiences based on real-world implementation challenges. For some, this communication void undermined their confidence in RDs and raised concerns about the feasibility of the AMR initiatives. The frustration expressed by these participants highlighted the critical importance of maintaining ongoing dialogue with community members to achieve long-term public health benefits.

"I'm saying we were left behind because ever since we finished with the co-creation event there has never been any communication back to answer until now." (Participant 3, FGD)

Participants also highlighted resource constraints as a barrier to implementing the co-created solutions. This concern appeared to underscore the practical challenges faced by community members in translating their newfound knowledge and enthusiasm into tangible action. Many participants expressed frustrations because they believed they possessed valuable information about AMR but lacked the means to effectively disseminate it. The absence of educational materials, visual aids, or other tangible resources made it difficult for them to substantiate their claims and convince others of the importance of responsible antibiotic use. This resource gap not only hampered their ability to conduct awareness campaigns but also undermined their credibility within the community. Some participants felt that without concrete or professional-looking materials, their messages were often met with scepticism or indifference. The lack of financial support for organizing events, printing materials, or even compensating volunteers' time further compounded these difficulties.

"For us we have learnt about this issue, but it is becoming difficult for people to believe us when we are sharing the messages with them because we have nothing to show them as evidence of what we are talking about, so it is becoming difficult to influence the change due to lack of the resources." (Participant 5, MCG, FGD)

## Discussion

This study assessed the process of implementing AMR RDs in Malawi, examining how participants experienced the engagement, what solutions they co-created, and what challenges they identified in implementation. While participants identified numerous benefits of the RDs approach, including improved AMR awareness, reported changes in attitudes and practices, and appreciation for the inclusive, participatory process, many also raised concerns related to implementation challenges, resource limitations, and the need for sustained engagement. Below, we discuss these findings in relation to existing literature.

The global threat of AMR is widely understood to demand efforts to increase public awareness and promote responsible antibiotic use across all sectors [20,21]. Concerns however still exists regarding the lack of public awareness about AMR, and the need for context-specific, culturally appropriate interventions [22]. Our findings suggest that participants across all stakeholder groups had limited prior knowledge of AMR and antibiotics consistent with broader patterns documented across sub-Saharan Africa [23], including among small-scale farmers in Blantyre where knowledge gaps about antibiotic use and AMR have been documented [24]. The RDs approach appeared to address this knowledge gap by providing a platform for knowledge exchange and collaborative learning that participants reported as enabling them to contribute to the co-production of AMR solutions. RDs therefore offer an innovative approach to engaging diverse stakeholders in co-creating solutions to address AMR [20].

These findings align with evidence from previous studies on participatory approaches, suggesting that RDs can potentially increase AMR knowledge and catalyze context-relevant solutions [25,26]. The Thailand RDs pilot reported similar outcomes, including improved AMR understanding among participants with limited prior knowledge and co-creation of context-specific solutions emphasizing local leadership [18]. However, notable differences emerged between the two settings. In Thailand, spontaneous community-led AMR initiatives were documented following the RDs, suggesting potential for sustained engagement beyond the formal project period [18]. No such spontaneous initiatives were observed in Malawi, which likely reflects important contextual differences between the two settings, including Thailand's relatively more developed healthcare infrastructure and stronger regulatory frameworks, greater resource availability for community-level action, and differences in existing community organization structures. Additionally, the COVID-19 pandemic disrupted the Malawi pilot, which may have dampened momentum for community-led action. This does not mean RDs are less feasible in countries similar to Malawi, but rather that additional support structures, sustained engagement, and dedicated resource allocation are essential to translate dialogue outcomes into community action in resource-constrained settings. Longitudinal impact assessments on RDs remain limited in both settings, highlighting an important knowledge gap [18].

The WHO Global Action Plan on AMR recommends that addressing AMR requires diverse stakeholders to collaborate closely [8]. A key success in the implementation of RDs in Malawi was the inclusive nature of the approach, bringing together healthcare providers, farmers, and community members to co-create solutions. Participants reported appreciating the value of diverse perspectives in co-creating solutions and expressed feeling empowered to contribute to address AMR. This aligns with other studies highlighting the importance of community engagement to empower communities to co-develop sustainable health interventions [27]. RDs participants also expressed recognition of the need for multi-sectoral collaboration to address AMR, and some reported initial small-scale actions to promote responsible antibiotic use following their participation. The need for sustained behaviour change was a key theme among participants in our study, who expressed seeing the opportunity to build on the momentum created by RDs and integrate some of the innovations into routine practices.

Despite participants expressed willingness to implement small-scale co-created interventions at individual level, our findings highlighted several perceived barriers to implementing AMR solutions, including weak regulations, lack of resources, and challenges in maintaining sustained engagement with communities. This echoes findings from other LMICs, where resource constraints often hinder health intervention implementation [28,29].

The participatory nature of RDs appeared to allow for the co-creation of multi-sectoral interventions, but participants expressed recognition of their limitations in implementing the co-created solutions or influencing meaningful change. To ensure translation and sustainability of the co-created interventions, participants emphasized the importance of strengthening health systems, regulatory frameworks for antibiotics use, financial support for awareness campaigns, and implementation of other interventions. As such, participants suggested that there was need for policy makers to enforce and support some of the co-created solutions to strengthen health systems and drug regulatory frameworks. This is in line with a policy analysis for developing countries that suggests that while National Action Plans exist, implementation gaps remain due to resource and capacity constraints [30]. In addition, participants also expressed challenges in translating knowledge into sustained behaviour change even at the individual level.

Poverty was highlighted as a significant barrier to reducing antibiotic use in livestock production or limiting sales of antibiotics at community level because most participants reported fearing economic losses. Participants appeared to perceive AMR as a distant threat compared to loss of livelihood and income. These findings reflect deeper structural and systemic issues that extend beyond individual behaviour. In many LMICs, informal health services, including drug shops and over-the-counter antibiotic dispensers, function as patches for structural deficits in formal healthcare systems [31,32]. When access to formal healthcare is constrained by distance, cost, and limited functional capacity, communities turn to these informal providers, for whom antibiotic sales represent a critical source of income [32]. This creates fundamental ethical tensions in AMR interventions: between personal economic interests of individuals whose livelihoods depend on selling or using antibiotics, and the wider collective interest in curbing AMR; and between addressing the immediate health needs of current populations who lack access to appropriate healthcare and preserving antibiotic effectiveness for future generations [32]. Our findings suggest that participatory approaches like RDs, while effective at raising awareness, cannot alone resolve these structural contradictions without concurrent investment in strengthening health systems, improving access to formal healthcare, and providing alternative livelihood options for those whose income depends on antibiotic sales. This aligns with other studies on behaviour change interventions, which highlight the complex interplay of individual, social, and environmental factors influencing health behaviours [33,34].

The participants in Malawi expressed a desire for feedback and continued engagement following the RDs. This desire stemmed from several factors. Having invested significant time and energy in the co-creation process, participants expected reciprocity and accountability regarding how their contributions would be used. The gap between the final co-creation event and any tangible implementation created uncertainty about the impact of their participation, with some expressing frustration at the perceived silence following the workshops. Furthermore, participants who had begun sharing AMR messages in their communities needed ongoing support, resources, and validation to sustain their advocacy efforts. This preference for ongoing communication and support has been highlighted in other studies on community-based interventions, emphasizing the importance of closing the feedback loop to maintain trust and sustain long-term impact [35].

It is important to acknowledge that AMR is widely recognized as a super-wicked problem, characterized by inherent complexity, multiple interrelated biological and social drivers, numerous stakeholders with competing interests, and limited time to act [36]. Addressing such problems requires not only participatory engagement but also systematic mapping of stakeholder interests, alignment, power dynamics, and transformation potential of the various actors involved [37]. RDs in Malawi represent a preliminary but important step in this process. Notably, the dialogues themselves generated the kind of critical reflexivity that is essential for sustained AMR action. Participants actively identified gaps in representation, including the absence of large-scale poultry farmers, demonstrating that the RD model can foster the self-awareness needed to iteratively expand and strengthen stakeholder engagement. Rather than signaling a limitation of the approach, this reflects the model working as intended: creating space for communities and stakeholders to collectively recognize who else needs to be at the table. Future iterations of AMR RDs can build on this reflexivity by incorporating more comprehensive and ongoing stakeholder analysis, ensuring that all relevant actors across the One Health spectrum, including commercial agricultural enterprises and regulatory bodies, are systematically identified and engaged over time.

While RDs appeared to be effective in increasing AMR awareness to co-create solutions and inspire action, sustained engagement appears to be important to ensure long-term impact. To ensure the sustainability of community-driven AMR solutions, long-term multi-sectoral stakeholder support is essential to maintain momentum and foster lasting behavior change. This could include local pilot projects to demonstrate the feasibility and impact of co-created solutions, providing valuable insights and data to inform larger-scale implementation. Future research should also investigate the sustained impact of RDs on antibiotic use practices; identify community-led strategies to fundraise and implement co-created solutions; and investigate how to effectively integrate RDs in other national AMR stewardship initiatives.

## Limitations

The findings of this study contribute to the growing body of evidence regarding participatory approaches to addressing AMR. However, several limitations should be considered when interpreting the results. The main limitation of this study was the focus on short-term outcomes immediately following the RDs workshops. While the findings highlight reported changes in attitudes and behaviours, the long-term sustainability of these changes remains uncertain as previously indicated. Furthermore, the evaluation focused exclusively on participant perspectives and did not systematically include the research team, facilitators, or other stakeholders involved in implementation. Multi-stakeholder evaluation would have provided more comprehensive insights into facilitation challenges, implementation barriers, and the feasibility of scaling up the approach. In addition, self-reported data is inherently subject to social desirability bias, particularly when discussing health behaviors, and should be interpreted with appropriate caution [38]. Follow-up studies tracking participants' sustained engagement with the co-created solutions would provide a more comprehensive understanding of the long-term impact of the RDs approach. Additionally, the absence of baseline data on participants' antibiotic use practices limits our ability to track actual behavior change. The co-created solutions were intentionally shaped by Malawi#39;s unique socio-economic context to ensure local relevance. While the participatory RDs approach itself is transferable, the specific solutions would require adaptation when applied in different socio-economic contexts, including other regions within Malawi.

We also acknowledge the importance of reflexivity in qualitative research. The lead researcher (HS) is a Chichewa-speaking Malawian researcher with extensive experience in qualitative health research in the study communities and was involved in the broader RDs implementation. While this insider perspective facilitated rapport, cultural understanding, and nuanced interpretation of participants' accounts, it may also have influenced how participants responded, particularly regarding perceived successes of the RDs process. Regular reflexive discussions within the research team, which included both Malawian and international researchers, helped to identify and mitigate potential biases in data collection and interpretation. Regarding triangulation, data triangulation was achieved through the use of multiple data collection methods (SSIs and FGDs) and inclusion of multiple participant groups (farmers, healthcare providers, male caregivers), allowing for comparison across perspectives and identification of convergent and divergent findings. Investigator triangulation was ensured through collaborative coding review and theme development involving multiple team members (DN, MP, JM, RP, EM). We acknowledge that methodological triangulation, for example incorporating observational data from the RDs themselves or document analysis of co-created outputs, would have further strengthened the findings.

## Conclusion

This study provides valuable insights into the process of implementing AMR RDs in Malawi, offering rich contextual understanding of implementation challenges and facilitators that can inform future participatory engagement initiatives. The RDs successfully brought together diverse stakeholders including farmers, healthcare providers, and male caregivers, facilitating collaborative learning, improved AMR awareness, and the co-creation of locally relevant solutions and policy priorities. Reported changes in attitudes and practices, particularly among healthcare workers, suggest that the participatory nature of RDs can influence not only community members but also those who play a crucial role in antibiotic stewardship.

However, our findings also highlight critical challenges that must be addressed for RDs to achieve sustained impact. Structural barriers including poverty, weak regulatory frameworks, and reliance on informal health services constrain the translation of improved awareness into lasting behaviour change. The absence of feedback mechanisms and resource support following the RDs undermined participants' confidence and ability to implement co-created solutions. Additionally, the one-off stakeholder mapping prior to the RDs resulted in the exclusion of key actors such as large-scale poultry farmers.

Considerations for future AMR RDs implementation include: incorporating comprehensive and ongoing stakeholder analysis to ensure inclusive engagement across the One Health spectrum; allocating resources for post-RDs follow-up and support to maintain momentum; integrating RDs outputs within existing health system structures and national AMR action plans; and establishing feedback mechanisms to sustain community engagement. Future research should investigate the long-term impact of RDs on antibiotic use practices and explore how participatory approaches can be effectively combined with structural interventions to address the systemic drivers of AMR in resource-limited settings.

## Supporting information

**S1 Text. Focus group discussion transcript — Farmers group.**
(DOCX)

**S2 Text. Focus group discussion transcript — Male caregivers group.**
(DOCX)

**S3 Text. Focus group discussion transcript — Healthcare providers group.**
(DOCX)

**S4 Text. Interview transcript — Farmer, participant 01.**
(DOCX)

**S5 Text. Interview transcript — Farmer, participant 02.**
(DOCX)

**S6 Text. Interview transcript — Farmer, participant 03.**
(DOCX)

**S7 Text. Interview transcript — Farmer, participant 04.**
(DOCX)

**S8 Text. Interview transcript — Farmer, participant 05.**
(DOCX)

**S9 Text. Interview transcript — Healthcare provider, participant 06.**
(DOCX)

**S10 Text. Interview transcript — Healthcare provider, participant 07.**
(DOCX)

**S11 Text. Interview transcript — Healthcare provider, participant 08.**
(DOCX)

**S12 Text. Interview transcript — Healthcare provider, participant 09.**
(DOCX)

**S13 Text. Interview transcript — Healthcare provider, participant 10.**
(DOCX)

**S14 Text. Interview transcript — Community caregiver, participant 11.**
(DOCX)

**S15 Text. Interview transcript — Community caregiver, participant 12.**
(DOCX)

**S16 Text. Interview transcript — Community caregiver, participant 13.**
(DOCX)

**S17 Text. Interview transcript — Community caregiver, participant 14.**
(DOCX)

**S18 Text. Interview transcript — Community caregiver, participant 15.**
(DOCX)

## Acknowledgments

We are grateful to all participants; poultry farmers, government and private pharmacists and prescribers, and male care-givers who generously shared their time, experiences, and insights during the RDs and subsequent evaluation. We extend our thanks to community leaders in areas of Chemusa, Ndirande, Chileka, Chilomoni in Blantyre for facilitating access and supporting the implementation of this work. We are grateful to the Malawi-Liverpool-Wellcome Programme, Ministry of Health, Ministry of Agriculture and Community Health Science Unit (CHISU) for institutional support.

## Author contributions

**Conceptualization:** MacWellings Phiri, Eleanor Macpherson, Deborah Nyirenda.

**Data curation:** Henry Sambakunsi.

**Formal analysis:** Henry Sambakunsi.

**Funding acquisition:** Jo Zaremba, Nicholas A Feasey, Eleanor Macpherson.

**Investigation:** Henry Sambakunsi.

**Methodology:** MacWellings Phiri, Eleanor Macpherson, Deborah Nyirenda.

**Project administration:** Henry Sambakunsi.

**Supervision:** Eleanor Macpherson, Deborah Nyirenda.

**Writing – original draft:** Henry Sambakunsi.

**Writing – review & editing:** MacWellings Phiri, Thomasena O'Byrne, John Mankhomwa, Raymond Pongolani, Jo Zaremba, Nicholas A Feasey, Eleanor Macpherson, Deborah Nyirenda.

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
