## [Decision Letter · Decision Letter 0]

6 Jan 2026

PGPH-D-25-03764

Co-creating community driven solutions and policy asks to address antimicrobial resistance through Responsive Dialogues: A qualitative evaluation from Malawi

Dear Dr. Sambakunsi,

Thank you for submitting your manuscript to PLOS Global Public Health. After careful consideration, we feel that it has merit but does not fully meet PLOS Global Public Health’s publication criteria as it currently stands. Therefore, we invite you to submit a revised version of the manuscript that addresses the points raised during the review process.

We look forward to receiving your revised manuscript.

Kind regards,

Bipin Adhikari, MBBS, DTM&H, MCTM, MPH, DPhil

Academic Editor

Journal Requirements:

Additional Editor Comments (if provided):

Title: consider modifying title to ensure it aligns with the most used terminologies.

Suggested: Co-creation of community-driven solutions and policy priorities to address antimicrobial resistance through Responsive Dialogues: A qualitative evaluation from Malawi

Introduction

Expand/acknowledge the multi-sectoral/multi-factorial contributions to AMR and why so more prominently in LMICs (e.g. regulatory, health system, formal/informal health services), this will naturally set a scene for Malawi

https://pubmed.ncbi.nlm.nih.gov/31799007/

Methods

Could you share semi-structured interview guide and FGD guide as supplementary Files.

Line 130: Add a line why you interviewed cattle and poultry farmers among animal sector?

Discussion

Line 397: “Poverty was highlighted as a significant barrier to reducing antibiotic use in livestock production or limiting sales of antibiotics at community level…..” Expand on it; these are structural issues; quite systemic and informal health care/services act as patches for structural deficit.

https://pubmed.ncbi.nlm.nih.gov/33975888/

There are ethical issues pertaining to this

https://pubmed.ncbi.nlm.nih.gov/38569658/

Line 402: Not clear. Explain why there was a desire for feedback and continued engagement following the RDs.

Line 404: While RDs appeared to be effective in increasing AMR awareness to co-create solutions and inspire reported behaviour changes, sustained engagement appear to be important to ensure long-term impact. This is indeed the most awaited benefit from any engagement activity but AMR is a unique and super-wicked problem, and therefore needs stakeholder analysis (interests, alignment, power, actors, policy and their transformation potential). Authors need to acknowledge this preliminary engagement process before being able to first understand the actors, their interests and second sustain the engagement.

https://pubmed.ncbi.nlm.nih.gov/41125295/

Reviewers' comments:

Reviewer's Responses to Questions

**Comments to the Author**

1. Does this manuscript meet PLOS Global Public Health’s publication criteria? Is the manuscript technically sound, and do the data support the conclusions? The manuscript must describe methodologically and ethically rigorous research with conclusions that are appropriately drawn based on the data presented.? Is the manuscript technically sound, and do the data support the conclusions? The manuscript must describe methodologically and ethically rigorous research with conclusions that are appropriately drawn based on the data presented.

Reviewer #1: Yes

Reviewer #2: Partly

2. Has the statistical analysis been performed appropriately and rigorously?

Reviewer #1: N/A

Reviewer #2: Yes

3. Have the authors made all data underlying the findings in their manuscript fully available (please refer to the Data Availability Statement at the start of the manuscript PDF file)?

The PLOS Data policy requires authors to make all data underlying the findings described in their manuscript fully available without restriction, with rare exception. The data should be provided as part of the manuscript or its supporting information, or deposited to a public repository. For example, in addition to summary statistics, the data points behind means, medians and variance measures should be available. If there are restrictions on publicly sharing data—e.g. participant privacy or use of data from a third party—those must be specified.requires authors to make all data underlying the findings described in their manuscript fully available without restriction, with rare exception. The data should be provided as part of the manuscript or its supporting information, or deposited to a public repository. For example, in addition to summary statistics, the data points behind means, medians and variance measures should be available. If there are restrictions on publicly sharing data—e.g. participant privacy or use of data from a third party—those must be specified.

Reviewer #1: No

Reviewer #2: Yes

4. Is the manuscript presented in an intelligible fashion and written in standard English?

Reviewer #1: Yes

Reviewer #2: Yes

Reviewer #1: Overall impression:

The manuscript can be confusing because it doesn’t explain the implementation of RDs through the conversation events well. There is no mention of how many participants were in the RDs, what interventions were co-created, or even how many CEs were conducted. I’m not sure what percentage of the CE participants you sampled from the evaluation, or how you sampled them. There are many unanswered questions that obscure the actual impact of the RD process.

The only reported results are improvements to awareness, knowledge, and attitudes, but these are from people you describe as not having previous AMR knowledge. The most significant result, reported behaviour change among health workers, is at the end of a change in attitudes section, and feels like an afterthought.

The conclusion of this manuscript, that this provides evidence that RDs can effectiveness support stakeholders to create interventions, is not supported by the results. I think that, instead of focusing on the usefulness and feasibility of RDs as an intervention by reporting improved knowledge and awareness , you focus more on a discussion of the process of RDs including discussing the co-creation process, and discussing the strengths, challenges, limitations, and the next steps, as a way to help people who plan on conducting similar co-creation activities in the future. This to me would be a more compelling piece.

Abstract

Line 27: Responsive Dialogues should include its corresponding abbreviation (RDs) after. You included it in line 31, so just remove that one as this is the first reference of RDs.

Introduction

Lines 49-50: It is stated that the WHO has identified AMR as a top global public health threat, but the citation is not WHO, instead linking back to an editorial. Please cite the relevant WHO document as a primary source, and include the year in the text.

Lines 75-84: This section is a little bit confusing. Was this the iterative consultation in Malawi as discussed in line 79, or is this a pilot after the fact? To me, it would make more sense to start with what RD is, the toolkit for public engagement, and then discuss how it was made iteratively in Thailand and Malawi in what year, and now this is a pilot of the RD in Malawi for AMR.

Line 96: Remove “what the toolkit calls”, so that the sentence is “Implementation of RBs involved Conversation Events, day-long meetings where the identified stakeholders and participants together explored AMR topics and co-created solutions to address AMR.” Just flows better.

Line 96: This is not labelled as the second phase in paragraph, but the first and third phase are explicitly labelled.

Line 99: The adaptations for COVID19 are not specified, is this masking, smaller groups, or more drastic like virtual meetings?

Line 100-102: The other first phase has its own paragraph, so it would make sense for phase two and three to have their own. Phase three is also lacking contextual information, suggest adding another sentence or two to describe this. Who was collating and synthesizing learning, the study team? People on the ground?

Methods

Line 113: Conversation events has been previously shorted to CE

Line 113: There is no abbreviation for Semi-structured interviews (SSI) which you use on line 115.

Line 129: Conversation events is an abbreviation

Line 131, 132: Responsive Dialogues are both abbreviations

Line 139-140: Please detail where FGDs are located and timelines, are they located at an individuals house or communal space? Was it an agreed time for everyone?

Line 147-154: Please specify how confidentiality was kept in FGDs, which inherently has people sharing information with others

Line 156: You have been using Semi-structured interviews, SSI, interviews, and now IDIs to describe your interviews. Please review the document and decide on one universal term for this and use throughout the document.

Line 156-157: Were people told not to self-identify in the interviews and that they were being recorded? If so, please include that in the ethics section. Also include who translated the IDIs and FGDs, and how many transcripts were randomly checked.

Line 161-164: Only one person coded all transcripts? That is a bit unusual

Line 214: Need a comma after farmers

Line 216-218: This sentence is more of a conclusion or discussion rather than a result. Suggest removal to a future section.

Line 226-228: This is a sentence for the discussion section not results.

Line 232-241: There are reported intentions but did anyone give examples of discussing with neighbours? Would be more impactful to be able to state it led to genuine examples of behaviour change similar to pharmacists in a previous section.

Discussion

Line 369-372: Its interesting in Thailand that they saw spontaneous AMR initiatives, and I’d assume they had the same time line as this project, months after the RDs. Please discuss more about why this didn’t happen in Malawi, is it more that the context doesn’t allow for it in Malawi versus Thailand, and then would RDs be less feasible in countries similar to Malawi?

I would also include the insights about not including large poultry farmers in this, as well as how this population was missed.

Reviewer #2: TITLE:

The title may be considered misleading. What exactly you do?

Did you co-create community-driven solutions and policy asks to address antimicrobial resistance through Responsive Dialogues through a qualitative evaluation in Malawi

Or you evaluated the implementation of the AMR Responsive Dialogues in Malawi

So far it seems you did the latter. Otherwise, I don’t see a framework/ model/ or list of the co-created community-driven solutions, and policy asks to address antimicrobial resistance.

ABSTRACT

The abstract needs some rewording.

Line 26 -Inappropriate antibiotic use due to low public knowledge and awareness.....

Line 27 - Responsive Dialogues (RD)

INTRODUCTION

Line 54-55- add reference

Line 74- abrupt subtitle

Line 75-102 - sounds more like methodology - it could go to the methods section or supplementary (a more detailed version)

METHODS:

General: There is need to quantify that the RD was an "AMR RD?" – for rigour (transparency).

RESULTS:

L168 and L172 – Short hand was already used for Responsive Dialogues, focus group discussions and key informant interviews.

DISCUSSION:

Line 353: What exactly did this study evaluate - feasibility ? these are are two different phenomenons. Evaluation is after implementation assuming feasibility was already determined????

Line 356: The following paragraph doesnt fit in with your previous paragraph: many also raised concerns related to implementation challenges,resource limitations, and they highlighted the need for sustained engagement.

After here the following paragraphs need major revisions- concordance and discordance of your findings with the current literature on AMR RD - or simply RDs.

On limitations: Consider adding something on Reflexivity, triangulation of researchers,

Conclusions: Amend accordingly

**Do you want your identity to be public for this peer review?** For information about this choice, including consent withdrawal, please see our Privacy Policy..

Reviewer #1: No

Reviewer #2: No

---

## [Editor Report · Decision Letter 1]

10 Mar 2026

Co-creating community driven solutions and policy asks to address antimicrobial resistance through Responsive Dialogues: A qualitative evaluation from Malawi

PGPH-D-25-03764R1

Dear Mr Sambakunsi,

We are pleased to inform you that your manuscript 'Co-creating community driven solutions and policy asks to address antimicrobial resistance through Responsive Dialogues: A qualitative evaluation from Malawi' has been provisionally accepted for publication in PLOS Global Public Health.

Best regards,

Bipin Adhikari, MBBS, DTM&H, MCTM, MPH, DPhil

Academic Editor